# “Can Do” vs. “Do Do” in Older Adults: A Cross-Sectional Analysis of Sensor-Derived Physical Activity Patterns

**DOI:** 10.3390/s23041879

**Published:** 2023-02-07

**Authors:** Michael Adams, Lara Carrascosa, Carl-Philipp Jansen, Yvonne Ritter, Michael Schwenk

**Affiliations:** 1Institute of Sports and Sports Sciences, Heidelberg University, 69120 Heidelberg, Germany; 2Human Performance Research Centre, Department of Sport Science, University of Konstanz, 78464 Konstanz, Germany; 3Department of Clinical Gerontology and Geriatric Rehabilitation, Robert Bosch Hospital, 70376 Stuttgart, Germany

**Keywords:** older adults, walking pattern, walking behavior, PAL, walking intensity, walking frequency, walking duration, physical activity, physical capacity, digital health monitoring

## Abstract

(1) Background: Identifying groups with a misaligned physical capacity (PC) and physical activity (PA) is potentially relevant for health promotion. Although an important health determinant, deeper knowledge of underlying walking behavior patterns in older adults is currently missing. We aim to identify specific PA signatures of misaligned groups and determine PA variables discriminating between groups. (2) Methods: In total, 294 community-dwelling older adults (≥70 years) were divided into four quadrants based on thresholds for PA (≥ or <5000 steps/day) and PC (≤ or >12 s, Timed Up and Go test). Kruskal–Wallis and effect sizes were calculated to compare quadrants’ PA variables and to determine the discriminative power of PA parameters on walking duration, frequency, and intensity. (3) Results: We identified quadrant-specific PA signatures. Compared with “can do–do do”, the “cannot do–do do” group performs shorter continuous and lower-intensity walks; the “can do–do not do” group takes fewer steps and walks with less intensity. The “cannot do–do not do” group presents lower values in all PA variables. “Walking duration greater or equal 3 METs” was the strongest discriminative PA variable. (4) Conclusion: We provide distinct PA signatures for four clinically different groups of older adults. Walking intensity is most useful to distinguish community-dwelling older adults, which is relevant for developing improved customized health promotion interventions.

## 1. Introduction

A large proportion of older adults do not reach physical activity recommendations despite sufficient physically capacity [1,2]. This misalignment of physical capacity (PC) and physical activity (PA) is a potentially useful starting point for the provision of health promotion interventions. The “can do–do do” or PC-PA quadrant concept by Koolen et al. [3] allows the identification of such misalignment, and with this, a better understanding of the discrepancies between PC and PA levels. For this purpose, subjects were classified binarily according to predefined thresholds for PC, in terms of what they “can do” or “cannot do”, and for PA, in terms of what they “do do” or “do not do”. The result is four quadrants to which a person is allocated: (1) “can do–do do” (high PC, high PA), (2) “can do–do not do” (high PC, low PA), (3) “cannot do–do do” (low PC, high PA), and (4) “Cannot do–do not do” (low PC, low PA). To date, the PC-PA quadrant concept has been applied in people with pulmonary diseases [3,4,5,6,7]. Here, the concept emerged as a promising approach to form clinically distinct groups and to contrast an individual’s ability to perform physical activities and their actual performance of PA in everyday life.

More profound knowledge of older adults’ specific PA patterns within each PC-PA category may be beneficial to achieve more accurate and personalized PA promotion among this population. Previous research suggests that walking is the most common PA among older people [8,9]. Therefore, walking PA patterns are the focus of this paper. Orendurff et al. [10] studied walking-related PA patterns of middle-aged adults for 14 days and found that 81% of their walking bouts were shorter than 70 s and that 20 s walking bouts were the most frequent walking bouts. Therefore, PA promotion needs to consider walking bout duration as a potent intervention aspect. The walking behavior of this population can be described as low-intensity interval activity or “training” with many stops and starts [10]. However, this does only apply to the overall mean. In the context of PA patterns, an in-depth characterization of walking-related behavior in subgroups, as carried out in the PC-PA quadrant concept, has yet to be done. One specific example would be the PA pattern of persons who are physically active, i.e., walk more than 5000 steps (“do do” category) [3,11,12], despite having a low physical capacity (“cannot do” category). To date, the specific PA patterns are often not investigated. Do older persons still perform longer walking bouts despite having a low PC or do they accumulate 5000 steps by multiple short walking bouts? In contrast, what is the PA pattern of older adults who have a high PC (“can do” category) but walk less than 5000 steps per day (“do not do” category)? Do they execute a few long walking bouts and remain sedentary for the rest of the day? To answer such questions, analyzing walking patterns with the PC-PA concept could be beneficial to understand walking behavior in the four quadrant populations, providing important information about individuals’ PC-PA misalignments, which could be targeted with clinical interventions.

This work aims to apply the PC-PA quadrant concept in healthy, community-dwelling older adults. The specific aim is to identify PC-PA quadrant-specific PA signatures.

## 2. Materials and Methods

### 2.1. Study Design and Participants

We used baseline data from the LiFE-is-LiFE trial, an intervention study conducted at Heidelberg University and Robert Bosch Hospital Stuttgart [13,14,15]. Information on sample size calculation can be found in the study protocol [15]. Participants were recruited between April 2018 and June 2019. All participants provided written informed consent to participate in this study. Further information on the recruitment process and inclusion criteria, study design, and ethics can be found in the study protocol [15].

### 2.2. Inclusion and Exclusion Criteria

To participate in the study subjects had to be 70 years or older; speak and read German; be able to walk 200 m with or without a walker; live at home or have “assisted living” without active assistance. Additionally, they had to match one of the following criteria within the last 12 months: one injurious fall, more than one non-injurious fall, or report of having perceived impaired balance and needing ≥12 s for completing the Timed Up and Go (TUG) test.

Subjects were excluded from participation if they exercised more than once a week in the previous 3 months; performed moderate to vigorous physical activity for longer than 150 min/week in the previous 3 months; had any serious medical condition, e.g., Parkinson’s disease, active cancer treatment, or acute depression [15]; suffered from moderate to severe cognitive impairment (Montreal Cognitive Assessment < 23); or actively participated in another scientific trial. Eligibility criteria were verified via telephone interviews and in-house screening. The reasons for this eligibility criteria were related to the intervention study focusing on older adults at risk of falling willing to participate in an exercise program.

### 2.3. Measurements

Socio-demographic and medical data (gender, age, educational level, height, weight, and living conditions) were recorded via interviews.

PC was assessed using the TUG test. The TUG test is a widely used performance-based assessment of functional mobility that contains several tasks that are highly relevant for PC [16]. It measures the time needed to stand up from a chair, walk three meters, turn 180 degrees, and walk back to the chair to sit down [17]. The test was performed twice, using the fastest one for our group classification analysis. PA was measured with an “activPAL4™ micro” accelerometer (PAL Technologies Ltd., Glasgow, Scotland). Subjects wore it on the anterior mid-thigh for 9 consecutive days. The sensor was attached on day one and taken off on day nine, meaning that seven full days (i.e., seven complete 24 h measurements between days two and eight) were analyzed. As day one and nine were incomplete, these were deleted from the analyses.

PA variables of walking intensity, duration, and frequency were assessed and classified as mean and maximum variables (Table 1). Mean variables were the extrapolated weekly average of the variable in question based on available data (if less than seven days of data were available), whereas maximum variables corresponded to the best values recorded during the days of accelerometer use.

### 2.4. PC-PA Quadrant Concept

After data collection, subjects were distributed into the four subgroups of the PC-PA quadrant concept from Koolen et al. (2019). The basis for group distribution was PA (“average steps per day”, steps) and PC (“TUG”, time). Individual results were plotted among both the x (PC) and y (PA) axes. Subjects were subdivided into the four quadrants based on the following requirements: (1) “can do–do do”, high PA (number of steps per day ≥ 5000 steps [3,12]) and high PC (TUG ≤ 12 s); (2) “cannot do–do not do”, low PA (number of steps per day < 5000 steps) and low PC (TUG > 12 s); (3) “can do–do not do”, high PA and low PC; and (4) “cannot do–do do”, low PA and high PC.

### 2.5. Data Analysis

Descriptive statistics were calculated for sample characteristics. Data normality was checked for each group using the Kolmogorov–Smirnov test. As data were not normally distributed, non-parametric tests were used. Baseline differences between the four groups were analyzed using the Kruskal–Wallis H test. The same test was used to compare PA characteristics among the four subgroups. Pearson’s Chi^2^ test was used for post hoc analyses. Effect sizes^®^ were calculated for each group comparison by dividing the Z-score by the square root of the number of the studied population *n* [r = z/√(n)] [18]. Values of 0.1–0.3 were interpreted as low effect sizes, 0.3–0.5 as moderate effect sizes, and values greater than 0.5 as high effect sizes. Statistical analyses were performed using SPSS (IBM Corp., Version 27.0. Armonk, NY, USA).

## 3. Results

A total of 294 older adults (mean age = 78.8 years; 72.8% women) were included in this analysis (Table 2). The application of the PC-PA quadrant concept (Figure 1) in this population resulted in the following distribution: “can do–do do” *n* = 81 (27.55%), “cannot do–do not do” *n* = 97 (32.99%), “can do–do not do” *n* = 45 (15.31%), and “cannot do–do do” *n* = 71 (24.15%). Participants in the four quadrants differed significantly (*p* < 0.01) in age, BMI, living situation, school grade, intrinsic motivation, TUG, and average steps per day, but not in gender distribution (Table 1).

### 3.1. Significance of Differences between the Four Quadrants in PA

Descriptive statistics and statistical significance of differences between the four quadrants in PA variables are displayed in Table 3. Significant differences between groups were found for all PA variables.

### 3.2. Effect Sizes

Effect sizes of the group differences are displayed in Figure 2, Figure 3 and Figure 4 and Appendix A.

### 3.3. Walking Intensity

The “p95 cadence” differed between all groups with a low effect size of 0.04 to 0.27 (mean = 0.14; SD = 0.09). In contrast, the effect sizes of the mean cadence ranged from <0.01 to 0.50 (mean = 0.32; SD = 0.22). In both variables, the smallest difference was found between the “can do–do do” and the “can do–do not do” quadrants. The largest differences were found between the “can do–do do” and the “cannot do–do not do” groups.

With respect to the “walking duration lower 3 METs”, the four quadrants differed with effect sizes of 0.05 to 0.66 (mean = 0.42; SD = 0.28). The smallest differences were found between the “can do–do not do” and the “cannot do–do not do” quadrants. The largest differences were found between the “can do–do not do” and the “cannot do–do do” groups.

In contrast, effect sizes of “walking duration greater or equal 3 METs” were between 0.07 and 0.88 (mean = 0.54; SD = 0.34). The smallest differences were found between the “can do–do do” and the “cannot do–do do” quadrants, while the largest effect sizes occurred between the “can do–do do” and the “cannot do not do” quadrants.

Considering the “percentage of walking duration equal or greater 3 METs” in relation to the total walking duration effect sizes ranged from 0.16 to 0.59 (mean = 0.32; SD = 0.17). The lowest effects appeared between the “can do–do not do” and the “cannot do–do do” groups. Large effect sizes occurred only between the “can do–do do” and “cannot do–do not do” groups.

### 3.4. Walking Duration

Effect sizes with respect to the “maximum number of continuous steps” ranged from 0.14 to 0.64 (mean = 0.32; SD = 0.18). The lowest effect sizes appeared between the “can do–do do” and “cannot do–do do” quadrants; the largest effect sizes occurred between the “can do–do do” and the “cannot do– do not do” quadrants.

For “maximum walking interval length”, effect sizes ranged from 0.09 to 0.55 (mean = 0.33; SD = 0.19). The lowest effect was seen between the “cannot do–do not do” and the “can do–do not do” quadrants. Again, the largest effect sizes occurred between the “can do–do do” and the “cannot do– donot do” quadrants.

With respect to the “average walking interval length”, effect sizes ranged from 0.11 to 0.54 (mean = 0.32; SD = 0.18). The smallest difference was found between the “cannot do–do not do” and “can do–do not do” quadrants. The largest difference was found between the “can do–do do” and “cannot do–do not do” quadrants.

In the “number of walking intervals”, effect sizes ranged from 0.02 to 0.72 (mean = 0.47; SD = 0.34). The smallest difference was seen between “can do–do not do” and “cannot do–do not do”, while the largest differences existed between “cannot do–do do” and “cannot do–do not do”.

### 3.5. Frequency

Considering “percentage of walking intervals greater 10 s”, effect sizes ranged from 0.05 to 0.57 (mean = 0.35; SD = 0.21). The smallest difference was found between the “can do–do do” and “cannot do–do do” quadrants. The largest difference existed between “cannot do–do do” and “can do–do not do”.

The “percentage of walking intervals greater 20 s” revealed effect sizes between 0.02 and 0.56 (mean = 0.29; SD = 0.22). The lowest effects appeared between the “can do–do do” and the “cannot do–do do” groups. Large effect sizes occurred only between the “can do–do do” and “cannot do–do not do” groups.

With respect to the “percentage of walking intervals greater 60 s” effect sizes ranged from 0.01 to 0.26 (mean = 0.12; SD = 0.11). The lowest effect sizes appeared between the “can do–do do” and “can do–do not do“ quadrants and the largest effect sizes occurred between the “can do–do do” and the “cannot do– do not do” quadrants.

## 4. Discussion

This explorative study presents the first application of the PC-PA quadrants concept in community-dwelling older adults, a population in which PA is particularly relevant to maintain health, independence, and quality of life [19,20,21]. It provides a distinct PA signature for each of the four groups based on their specific underlying PA patterns. We identified walking intensity to be the strongest PA variable to distinguish older adults of different PA and PC levels.

### 4.1. Underlying Physical Activity Patterns of the Four PC-PA Groups

#### 4.1.1. Can Do–Do Do vs. Cannot Do–Do Do

As expected, the participants in the “can do–do do” group took the most steps (mean: 8047 steps per day) of all groups. There was no significant difference compared to the “cannot do–do do” group (7767 steps per day), meaning that both groups fulfil the recommendations for older adults suggesting 7100 steps per day [12] as an optimal threshold. The “cannot do–do do” group results show that 24.2% percent of older adults can maintain a high level of PA, despite having a low capacity. The overall PA level of this group may be more strongly related to behavioral aspects and routines than to raw capacity. As has been shown, the relationship between capacity and performance is not linear [9,22].

While not differing in total steps, our results highlight that both “do do” groups differed in specific PA patterns. First, the maximum number of continuous steps was higher in the “can do–do do” group, indicating that maximum values are related to individuals’ capacity. Restrictions in more demanding mobility tasks such as walking longer distances often indicate declining mobility [23]. Previous studies demonstrated strong relationships between the distance older people can walk and health outcomes such as cardiovascular risk factors, future mobility limitations, and mortality [24]. Moreover, persons with limited maximum walking distance are restricted from, e.g., performing long walks without seating areas or participating in social events [23].

Second, the “can do–do do” group had a significantly longer duration of walking intensities with equal or greater than 3 MET. This finding comes with high relevance for older adults’ health considering that the intensity of walking is more closely related to all-cause mortality than the duration of daily walking [25]. Current studies indicate that increasing the duration of intense walking is a promising approach to increasing relevant health outcomes. For example, results from a large survey indicated that moderate to vigorous leisure-time walking supports mental health and health perceptions in older adults [26] and is associated with decreased risk of developing type 2 diabetes [27]. This highlights the clinical relevance of detecting the “cannot do–do do” group and supporting them in improving specific walking parameters related to duration and intensity.

In summary, the “can do–do do” group is characterized by taking more steps in a row and walking longer at a higher intensity than the other PC-PA groups. The “cannot do–do do” group takes a comparable amount of steps but performs shorter continuous walks and walks with a lower intensity. This discrepancy in walking behavior is probably of high clinical relevance, emphasizing the value of this classification scheme.

#### 4.1.2. Can Do–Do Do vs. Can Do–Do Not Do

Despite the “can do–do not do” group’s high PC, the average number of steps in this group was low (3583 steps/day) compared with the “can do–do do” group (8387 steps/day). This contradicts results [28] suggesting that functional fitness is associated with high levels of walking (at least 6500 steps per day).

In terms of walking intensity and duration, we found significantly higher values for the “can do–do do” group compared with the “can do–do not do” group in all PA variables except “P95 cadence”. This means that the “can do–do not do” group takes rapid steps at the same peak cadence as more active populations, but has a significantly lower mean cadence in everyday life. This is in line with the classification as a group of a high PC but low PA. Reasons for the low level of PA may lie in a lower motivation for PA. There may also be social, organizational, and environmental aspects [29]; however, further research is needed to understand this misalignment of PC and PA.

Considering walking frequency, the “can do–do not do” group has a significantly higher proportion of very short (<10 s: 71.69%) and short (<20 s: 88.65%) walking bouts than the “can do–do do” group (<10 s: 64.08%; <20 s: 84.96%), but not moderate (<60 s: 97.02%) bouts. This shows that the “can do–do not do” group has a higher proportion of walking intervals of 1 to 20 s in daily life, while the “can do–do do” group has a higher percentage of walking intervals of 20–60 s. Orendurff et al. [10] investigated walking behavior in a sample of healthy adults (mean age: 36.3 years; SD: 14.8 years); they found most typical activities to be composed of a small number of continuous steps, and found only 46.1% of the walking intervals to be shorter than 20 s. Considering that older adults walk slower than younger populations, one might expect longer walking intervals that are needed to manage daily tasks and chores. Therefore, another explanation might be that the “can do–do not do” group takes significantly more breaks to complete longer walking distances.

In summary, the “can do–do not do” group takes significantly fewer steps per day, walks with lower intensity, and avoids distances longer than 20 s. The largest effect sizes comparing the “can do–do do” and “can do–do not do” groups occurred in “walking duration greater or equal 3 METs” (ES = 0.69). Since this group has a sufficient PC for brisk walking, there might be a particularly high potential to increase the duration of daily walking, intense walking, the number of long-distance walks, and related health outcomes [25,26,27].

#### 4.1.3. Can Do–Do Do vs. Cannot Do–Do Not Do

Compared with the “can do–do do” group the “cannot do–do not do“ group had significantly lower values in all PA variables related to walking duration and intensity. As expected, the “cannot do–do not do” group had a significantly higher proportion of walking intervals of less than 10, 20, and 60 s compared with the “can do–do do” group. The primary reasons for this may lie in lower PC. However, the average “P95 cadence” in this group was still above 100 steps per minute, a heuristic value for brisk walking with an energy expenditure of 3 MET or more [30]. This is consistent with our finding that this group sometimes walks at higher intensity, but for a distinctly shorter duration than all other groups.

The number of “maximum continuous steps” of the “cannot do–do not do” group was particularly low (mean: 606.6 steps; SD: 400.8 steps) compared with all other groups. Given the average normative step length of older adults is 57 cm for women and 66 cm for men [31], we can assume that a large proportion of this sample never walked 400 meters without stopping. Despite the fact that we could not measure step length, our findings might be clinically meaningful, as those unable to complete a 400 m walk test without stopping are at increased risk of early death, cardiovascular diseases, future mobility limitation, and walking disability [24].

In summary, the “cannot do–do not do” group takes significantly fewer steps per day, walks with lower intensity, and has a higher proportion of walks up to 60 s than the “can do–do do” group. This is in line with the classification as a group of low PC and low PA.

### 4.2. Most Relevant PA Variables for Discriminating the Four Groups (Largest Effect Sizes)

#### 4.2.1. Intensity

Overall, the walking intensity had the greatest discriminative power: “Walking duration greater or equal 3 METs” (mean ES = 0.54, range = 0.07–0.88) showed the largest effect sizes for group comparisons of all PA variables. The high clinical relevance of walking intensity has been discussed above [25,26,27]. Accordingly, this variable is particularly interesting when it comes to differentiating community-dwelling older adults based on their walking behavior.

#### 4.2.2. Duration

In terms of walking duration, “number of walking intervals” (mean ES = 0.47; range = 0.02–0.72) was the strongest overall discriminator. The “do not do” groups executed relatively more short and moderate intervals, while the “do do” groups presented higher numbers of walking intervals. The “do do” groups not only had a higher proportion of long-distance walks, but walked more often for short, moderate, and long walking durations. Therefore, “number of walking intervals” is the strongest discriminator regarding walking duration variables.

#### 4.2.3. Frequency

The “do not do” groups have a higher proportion of short and moderate walking intervals than the “do do” groups relative to their total number of walking intervals. The effect size for the percentage of short intervals (<10 s; mean ES = 0.35; range = 0.05–0.57) is larger than that for moderate (<20 s; mean ES = 0.28; range = 0.02–0.45) and long intervals (<60 s; mean ES = 0.10; range = 0.01–0.21). This might be because the “do not do” groups compensate for long distances by taking more breaks and walking in shorter walking intervals. Accordingly, differences between the PC-PA groups arise more clearly in the proportion of short to moderate distances than in long distances. There is probably a threshold above 60 s at which primarily “do do” groups perform walking intervals, and even discrimination within the “do do” groups might be possible. However, since we did not analyze walking intervals longer than 60 s, further studies are needed to confirm this.

### 4.3. Limitations

This exploratory study had some limitations. Using steps per day and the TUG as discriminative measures to define thresholds for PA and PC levels, we referred to established measures for our target populations. However, several thresholds have been identified in the research [32,33,34] and there is a debate regarding the most sufficient and meaningful cut-off values. Based on the thresholds we applied, several significant differences were identified in the PA variables. Of course, our results are related to the cut-off values we applied. The application of different cut-off values might lead to different group signatures. However, the fact that we were able to identify these group differences supports the significance of the thresholds applied in this study and the informative value of our investigation. The sample distribution into physically active “do do’s” and inactive “do not do’s” was based on the individuals’ average steps per day. We acknowledge that participants may have been engaged in other PA activities (e.g., swimming, cycling) that the accelerometer did not account for. This may have biased the group distribution. Furthermore, we explored only PA variables to describe and explain different levels of PA and PC. However, psychosocial and behavioral aspects are just as important for understanding and targeting patients with low daily PA [35,36]. Due to the cross-sectional nature of our investigation, it is not possible to investigate causal relationships between PC and PA; this should be part of future investigations to allow and improve individualized intervention recommendations. Finally, we used a selected sample from an intervention study with specific inclusion criteria. This limits the external validity of our results and future studies with non-selected samples should confirm them.

## 5. Conclusions

It is of clinical importance to identify persons with misaligned PA and PC. There is great potential for improvement through interventions, especially in the “can do–do not do” group. For this group, motivational interventions to increase PA might be helpful, emphasizing walking intensity because these persons are capable of brisk walking. In the “cannot do–do do” group, PC should be improved to allow those persons to perform longer walking episodes with higher intensity. Looking at the group with aligned PC and PA, the “cannot do–do not do” persons should aim for both better PC and more overall PA. This is important to underpin changes in PA behavior with improvements in PC, especially to avoid excessive demands on PC which could come with a higher risk of falling or other adverse events. Wearable motion sensors will most likely become an important pillar for health promotion interventions for older adults, e.g., as part of a continuous health monitoring and diagnostic device [37]. Researchers working on technology-assisted PA interventions could make use of these group signatures to apply highly specific measures and training for older adults, a population with a particularly high degree of heterogeneity in terms of functional capacities.

## Figures and Tables

**Figure 1 sensors-23-01879-f001:**
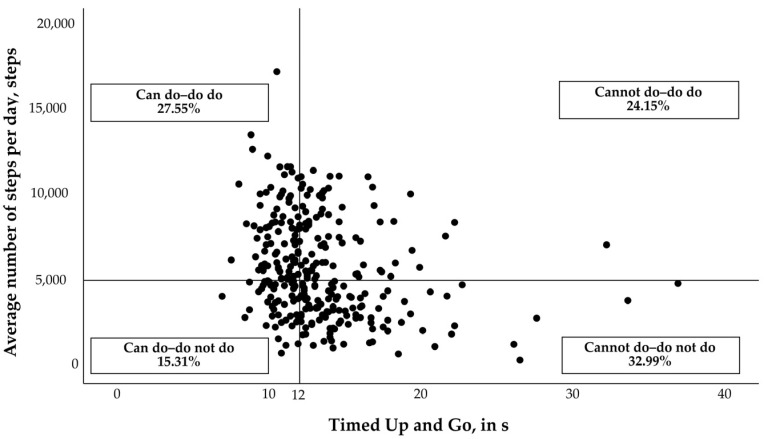
Graphical representation of the group division based on the physical capacity-physical activity (PC-PA) quadrant concept.

**Figure 2 sensors-23-01879-f002:**
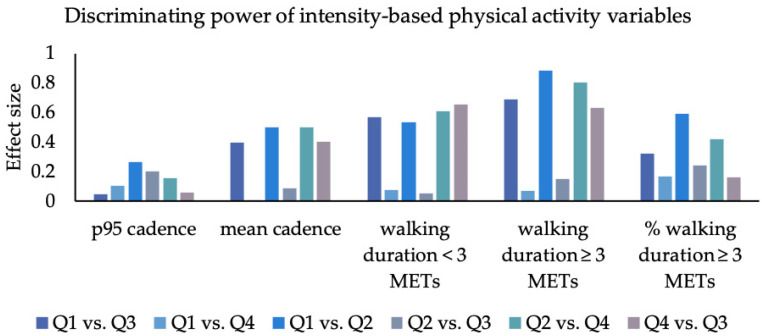
Effect size for each group comparison for intensity-based PA variables: Q1: 1. quadrant (can do–do do), Q2: 2. quadrant (cannot do–do not do), Q3: 3. quadrant (can do–do not do), Q4: 4. quadrant (cannot do–do do); P95 cadence (in steps per minute), mean cadence (in steps per min), walking duration lower 3 METs (in min), walking duration greater equal 3 METs (in min), percentage walking duration greater equal 3 METS (in min), percentage walking duration greater or equal 3 METs (in min).

**Figure 3 sensors-23-01879-f003:**
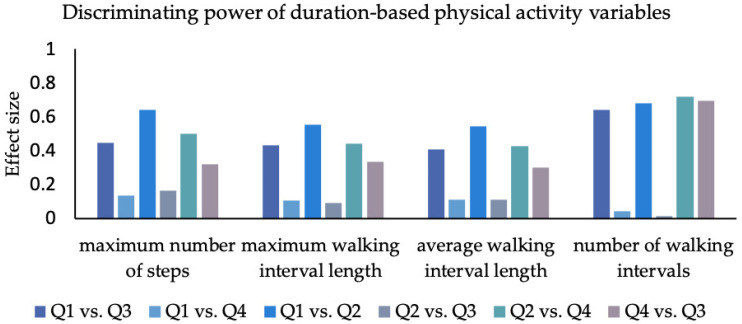
The effect size for each group comparison for duration-based PA variables. Q1: 1. quadrant (can do–do do), Q2: 2. quadrant (cannot do–do not do), Q3: 3. quadrant (can do–do not do), Q4: 4. quadrant (cannot do–do do); maximum continuous number of steps, maximum walking interval length (in s), average walking interval (in s), number of walking intervals.

**Figure 4 sensors-23-01879-f004:**
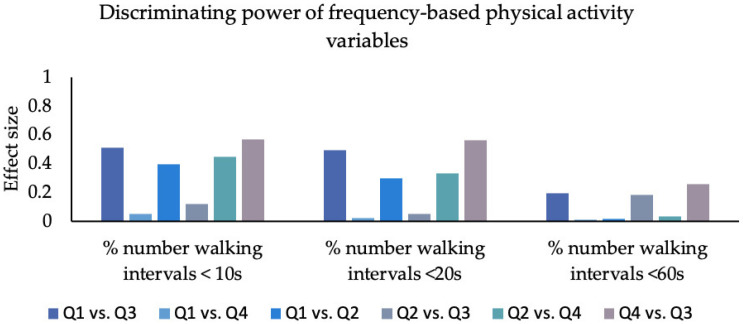
The effect size for each group comparison for frequency-based PA variables. Q1: 1. quadrant (can do–do do), Q2: 2. quadrant (cannot do–do not do), Q3: 3. quadrant (can do–do not do), Q4: 4. quadrant (cannot do–do do); percentage (%) number of walking intervals < 10 s, percentage (%) number of walking intervals length < 20 s, percentage number of walking intervals length < 60 s.

**Table 1 sensors-23-01879-t001:** PA variables.

PA Property	PA Variable
Intensity	“Mean cadence”, steps per minute “p95 cadence”, steps per minute “Walking duration lower 3 METs”, minutes “Walking duration greater equal 3 METs”, minutes “Percentage walking duration greater equal 3 METs”, %
Duration	“Maximum continuous number of steps”, number of steps “Steps per day”, averaged number of steps “Maximum walking interval length”, seconds “Average walking interval length”, seconds
Frequency	“% of walking intervals < 10 s”, % “% of walking intervals < 20 s”, % “% of walking intervals < 60 s”, %

METs: metabolic units; p95: 95th percentile.

**Table 2 sensors-23-01879-t002:** Descriptive characteristics of the four quadrants.

	Can Do–Do Do		Can Do–Do Not Do		Cannot Do–Do Do		Cannot Do–Do Not Do		*p* *
	Mean	SD	Mean	SD	Mean	SD	Mean	SD	
Subjects	81	-	45	-	71	-	97	-	-
% female	71.6	-	62.2	-	78.9	-	74.2	-	0.26 **
Age, years	77.3	4.7	76.1	4.7	79.9	5.4	80.5	5.3	<0.001
BMI, kg/m^2^	26.1	4.0	27.6	4.9	25.8	4.0	28.9	5.6	<0.001
TUG, s	10.6	1.0	10.5	1.1	14.7	3.2	15.8	4.3	<0.001
Steps per day, steps	8047	2338	3583	1161	7767	1976	3077	1107	<0.001
Number of walking intervals < 10 s	279.2	82.2	184.7	57.7	290.2	101.6	178.2	60.6	<0.001
Number of walking intervals < 20 s	371.3	108.1	229.9	72.3	390.3	128.9	226.7	74.9	<0.001
Number of walking intervals < 60 s	424.5	125.8	252.0	78.7	445.4	141.9	248.2	80.6	<0.001

* Calculated using Kruskal–Wallis H test ** Calculated using Pearson’Chi^2^ test.

**Table 3 sensors-23-01879-t003:** Comparison of PA variables.

	Can Do–Do Do (Q1)		Can Do–Do Not Do (Q2)		Cannot Do–Do Do (Q3)		Cannot Do–Do Not Do (Q4)		*p*-Value *
PA Variable	Mean	SD	Mean	SD	Mean	SD	Mean	SD	
P95 cadence (in steps per min)	108.67	7.84	107.64	8.23	107.24	8.44	104.31	8.62	0.003
Mean cadence (in steps per min)	67.68	4.81	63.06	4.91	68.03	5.39	61.73	5.68	<0.001
Mean walking duration per day (in min)	105.61	28.13	51.55	15.42	105.9	25.43	47.54	14.92	<0.001
Walking duration lower 3 METs (in min)	48.95	17.50	28.43	10.00	53.82	21.21	30.32	10.77	<0.001
Walking duration greater equal 3 METs (in min)	56.66	19.96	23.13	9.97	52.12	18.14	17.23	10.06	<0.001
Percentage walking duration greater equal 3 METs (%)	53.66	11.63	44.13	13.08	49.51	13.80	34.30	16.50	<0.001
Maximum continuous number of steps	1585.3	1009.6	868.36	816.6	1209.5	674.8	606.6	400.8	<0.001
Maximum walking interval length (in s)	15.16	9.86	8.50	5.95	12.19	6.27	7.09	4.58	<0.001
Average walking interval length (in s)	15.04	3.58	12.03	2.55	14.70	4.31	11.59	3.58	<0.001
Number of walking intervals	437.01	126.47	259.13	79.10	457.83	140.87	254.25	80.61	<0.001
Percentage of walking intervals < 10 s (%)	64.08	5.69	71.69	6.13	62.80	6.56	69.83	7.98	<0.001
Percentage of walking intervals < 20 s (%)	84.96	3.95	88.65	3.27	84.70	4.90	88.74	5.29	<0.001
Percentage of walking intervals < 60 s (%)	96.91	1.71	97.02	1.42	96.91	2.00	97.38	1.98	0.021

* Calculated using the Kruskal–Wallis H test; s = seconds; min = minutes; ES = effect sizes (largest ES from a comparison among the four subgroups).

## Data Availability

The data presented in this study are available on request from the corresponding author.

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
