# Peer review of "“Can Do” vs. “Do Do” in Older Adults: A Cross-Sectional Analysis of Sensor-Derived Physical Activity Patterns"

_sensors, 2023, doi:10.3390/s23041879_

Round 1

Reviewer 1 Report

Line 41; please provide detials of PC-PA quadrant concept. 

Line 61; why did the authors categorised by walking > 5000 steps? 

There are several investigations of physical capacity eg., TUG, sit to stand, gait speed, why did the authors use TUG? please provide the rationale of it. 

Line 104; please provide the rationale why the authors ask the participants to wear accelerometer for 9 days.

Please provide the how to calculate sample size; why it was 294 older people. 

Please revised the figures, these are difficult to read and understand.

Author Response

"Thanks for the reviewer's comments. Please find the authors' responses in the attachment."

Reviewer 2 Report

This is a paper likely to be of most interest to researchers. The data are presented clearly as is the comparison of PA capacity to performance.  However, the presentation lacks context in several instances and would be much improved if the context were added:

The reasons for the inclusion and exclusion criteria should be provided.  Without them, they appear to be incongruent with the purpose of the analysis.  This includes truncation of the sample, removing both high and low performers.

A focus on walking is acceptable but to what extent did some of the participants engage in other PA, particularly bicycling, group exercise, and so forth.  Would they have been in different categories if other types of PA were considered?

Importantly, the "can't do" label was applied to a substantial number of participants that actually "do do".  If they really cannot do PA, then how is it that they are doing quite a lot? The authors do provide some speculation on this but it does seriously undermine these descriptive phrases and raises questions about their usefulness as currently defined.

Author Response

(The authors gave the same response as above.)

Round 2

Reviewer 1 Report

Please check the quotation marks in the manuscript. 

e.g., line 26:  do-do do”, the „Can’t do-do do“

line 28: „Walking duration greater or equal 3 METs“

Author Response

We thank the reviewer for this comment. We have carefully checked and corrected the quotation marks.